# Metabolomic Analysis Reveals Distinct Profiles in the Plasma and Urine Associated with IgE Reactions in Childhood Asthma

**DOI:** 10.3390/jcm9030887

**Published:** 2020-03-24

**Authors:** Chih-Yung Chiu, Mei-Ling Cheng, Meng-Han Chiang, Chia-Jung Wang, Ming-Han Tsai, Gigin Lin

**Affiliations:** 1Division of Pediatric Pulmonology, Department of Pediatrics, Chang Gung Memorial Hospital at Linkou, and Chang Gung University, Taoyuan 333, Taiwan; jrbx@cgmh.org.tw; 2Clinical Metabolomics Core Laboratory, Chang Gung Memorial Hospital at Linkou, Taoyuan 333, Taiwan; chengm@gap.cgu.edu.tw; 3Department of Biomedical Sciences, and Metabolomics Core Laboratory, Healthy Aging Research Center, College of Medicine, Chang Gung University, Taoyuan 333, Taiwan; 4Department of Medical Imaging and Intervention, Imaging Core Laboratory, Institute for Radiological Research, and Clinical Metabolomics Core Laboratory, Chang Gung Memorial Hospital at Linkou, College of Medicine, Chang Gung University, Taoyuan 333, Taiwan; 0914.neo@gmail.com; 5Department of Pediatrics, Chang Gung Memorial Hospital at Keelung, College of Medicine, Chang Gung University, Taoyuan 333, Taiwan; drtsai1208@gmail.com

**Keywords:** allergic sensitization, asthma, branched-chain amino acid, metabolomics, urine

## Abstract

Several metabolomics studies have identified altered metabolic pathways that are related to asthma. However, an integrative analysis of the metabolic responses across blood and urine for a comprehensive framework of asthma in early childhood remains lacking. Fifty-four age-matched children with asthma (n = 28) and healthy controls (n = 26) were enrolled. Metabolome analysis of the plasma and urine samples was performed using ^1^H-nuclear magnetic resonance (NMR) spectroscopy coupled with partial least-squares discriminant analysis (PLS-DA). Integrated analysis of blood and urine metabolic profiling related to IgE reactions for childhood asthma was investigated. A significantly higher plasma histidine level was found, in parallel with lower urinary 1-methylnicotinamide and trimethylamine N-oxide (TMAO) levels, in children with asthma compared to healthy controls. Compared to children without allergic sensitization, 11 (92%) plasma metabolites and 8 (80%) urinary metabolites were found to be significantly different in children with IgE and food sensitization respectively. There were significant correlations between the plasma 3-hydroxybutyric acid and excreted volumes of the hydroxy acids, which were strongly correlated to plasma leucine and valine levels. Urine N-phenylacetylglycine, a microbial-host co-metabolite, was strongly correlated with total serum and food allergen-specific IgE levels. Plasma pyruvate and urine valine, leucine, and isoleucine degradation metabolisms were significantly associated with allergic sensitization for childhood asthma. In conclusion, blood and urine metabolome reflect different metabolic pathways in allergic reactions. Plasma pyruvate metabolism to acetic acid appears to be associated with serum IgE production, whereas urine branched-chain amino acid metabolism primarily reflects food allergic reactions against allergies.

## 1. Introduction

Asthma and other allergic diseases are conditions that result from the interactions between multiple genetic factors and various environmental stimuli [1]. Allergen exposure with IgE sensitization is a key environmental determinant for the onset of asthma, which involves a wide range of cellular and molecular mediators [2]. Clinically, metabolomics has uncovered the metabolic signatures of asthma and has provided novel insights into asthma profiling at the molecular level [3]. Metabolites that are involved in allergic reactions in response to allergen exposure have the potential to serve as specific biomarkers for asthma. However, only a few studies have addressed the relationship between metabolic profiles and allergic sensitization in relation to asthma in early childhood.

Metabolites are very responsive to dietary exposure as diet is an important source of metabolite variation and can induce specific biological activity. Plasma and urine metabolomics are commonly used to evaluate the molecular mechanisms that are responsible for human diseases [4]. Urine typically contains metabolic breakdown products from a wide range of foods, drinks, and drugs. Blood, on the other hand, contains metabolically active compounds, and lipid-soluble metabolites that are not present in urine. Several studies have identified altered metabolic pathways in association with asthma [5,6]. However, an integrative analysis of metabolic responses across blood and urine to elucidate childhood asthma has not yet been fully attempted.

We hypothesized that integrated analyses of metabolic profiles in blood and urine would provide more detailed information, highlighting the complexity of childhood asthma. Furthermore, a comprehensive understanding of the complexities of metabolic pathways involved in asthma may determine potential pathways involved in mediating the allergic responses to allergens and identify metabolite biomarkers potentially important for the treatment of asthma. The major aim of this study was to determine the plasma and urine metabolic profiles using ^1^H-NMR spectroscopy in patients with childhood asthma and healthy controls. The relationship between metabolites and atopic indices including IgE and allergen sensitization were assessed, and their relevance to the risk of asthma was examined.

## 2. Methods

### 2.1. Study Population

A prospective cross-sectional controlled study was conducted to investigate the metabolomic profiles of plasma and urine in children aged between 3 and 5 years old who diagnosed with asthma alone and healthy controls. The diagnosis of asthma was evaluated and physician-diagnosed at the outpatient clinics, based on the guidelines of the Global Initiative for Asthma [7]. Healthy children without a history of atopic conditions or infections were enrolled as controls. Information regarding demographic data, family atopy history, passive smoking, and household income related to atopic diseases were collected. This study was approved by the ethics committee of Chang Gung Memory Hospital (No. 103-1752A3). Written informed consent was obtained from the parents or guardians of all study subjects.

### 2.2. Measurement of Serum and Allergen-Specific IgE Levels

The serum samples of enrolled subjects were collected at outpatient clinics in the morning and immediately stored at −80 °C in aliquots until required. The total serum and allergen-specific serum IgE levels were examined as described in our previous study [8]. The total serum IgE level was measured by ImmunoCAP (Phadia, Uppsala, Sweden), and the specific IgE levels to food (egg white and cow’s milk) and inhalant (Dermatophagoides pteronyssinus and Dermatophagoides farinae) allergens were determined using a commercial assay for IgE (ImmunoCAP Phadiatop Infant; Phadia) [9]. Allergen-specific IgE levels ≥ 0.35 kU/L was defined as positive and indicative of allergic sensitization [10].

### 2.3. Plasma and Urine Sample Preparation

Spot plasma and urine samples prior to spectrum acquisition were prepared as described previously [11,12]. In brief, 500 μL of plasma was mixed with 500 μL phosphate buffer in deuterium water containing 0.08% 3-(trimethylsilyl)-propionic-2,2,3,3-d_4_ acid sodium salt (TSP), whereas 900 μL of urine was mixed with 100 μL of 1.5 M phosphate buffer in deuterium water containing 0.04% TSP as an internal chemical shift reference standard. The samples were vortexed for 20 s and centrifuged at 12000g at 4 °C for 30 min. Lastly, 600 μL supernatant was transferred to a 5-mm NMR tube for analysis.

### 2.4. ^1^H–Nuclear Magnetic Resonance (NMR) Spectroscopy

^1^H-NMR spectroscopy was performed using a Bruker Avance 600 MHz spectrometer (Bruker-Biospin GmbH, Karlsruhe, Germany) located at the Chang Gung Healthy Aging Research Center, Taiwan. A total of 64 scans were collected for NMR spectra into 64 K computer data points with a spectral width of 10,000 Hz (10 ppm). Prior to zero-filled Fourier transformation at an exponential line broadening of 0.3 Hz, 1D spectra were applied. The acquired ^1^H-NMR spectra were then manually phased, baseline-corrected, and referenced the chemical shift to TSP (δ 0.0 ppm) using TopSpin 3.2 software (Bruker BioSpin, Rheinstetten, Germany).

### 2.5. NMR Data Processing and Analysis

The raw ^1^H-NMR spectra were imported into NMRProcFlow, an open source software providing comprehensive tools for spectra processing, ppm calibration, baseline correction, alignment, spectra bucketing, and data normalization [13]. Least-squares algorithm and parametric time warping were used to adjust the misalignment for ^1^H-NMR spectra alignment. Spectra bucketing was performed using the method of intelligent bucketing and variable size bucketing [14]. Metabolites were identified by using the Chenomx NMR Suite 8.1 software (Chenomx Inc., Edmonton AB, Canada). Urine spectra were specifically normalized to the integral of creatinine peak at δ 3.045 ppm to compensate the differences in urinary concentration.

As previously established NMR data analysis methods [12], the normalized ^1^H-NMR spectra data were transformed using generalized log transformation (glog) and uploaded to MetaboAnalyst 4.0 (http://www.metaboanalyst.ca) to identify metabolites used for discrimination between the groups using partial least squares-discriminant analysis (PLS-DA). Spectral variables were mean-centered and scaled using Pareto scaling. A further 10-fold internal cross-validation was performed to assess the quality of statistical models using the diagnostic measures R^2^ and Q^2^ [15]. Metabolites with a variable importance in projection (VIP) score ≥ 1.0 or *p*-value < 0.05 were selected. The Kyoto Encyclopedia of Genes and Genomes database (KEGG) was employed to analyze the functional metabolic pathways.

### 2.6. Statistical Analysis

The baseline characteristics between children with asthma and healthy controls were compared using univariate parametric and non-parametric tests as appropriate, including Student’s t-test, Mann-Whitney test, χ^2^, and Fisher’s exact test. The differences in metabolites between the two groups were assessed using the Mann-Whitney test with the MetaboAnalyst web server. A false discovery rate (FDR) of 5% was applied to correct for multiple tests. The correlation coefficients between plasma and urine metabolites, and their relevance to allergen-specific IgE levels were assessed using Spearman’s correlation test in R software (Lucent Technologies, NJ, USA, version 3.5.3). Statistical analysis was performed using the Statistical Program for Social Sciences software v.20.0 (IBM SPSS Statistics for Windows, Armonk, NY, USA). All tests were two-tailed, and a *p*-value < 0.05 was considered statistically significant.

## 3. Results

### 3.1. Population Characteristics

A total of 54 children (28 asthmatics and 26 healthy controls) were prospectively enrolled in this study. The difference in the baseline characteristics between children with asthma and healthy controls is shown in Table 1. Atopic indices including total serum IgE levels and allergen-specific IgE levels to *D. pteronyssinus* and *D. farinae*, but not for egg white or cow’s milk were significantly higher in children with asthma than in the healthy controls (*p* = 0.001). There were no differences in sex, maternal age, maternal atopy, passive smoking, and household income.

### 3.2. Metabolites Sets Categorized by Asthma Outcome and Mite, Food, and IgE Sensitization

^1^H-NMR spectra obtained from plasma and urine corresponded to 34 and 44 known metabolites, respectively. The metabolites that contributed toward distinguishing between the groups of each dataset were identified using PLS-DA. The PLS-DA cross-validation and permutation test for distinguishing between groups are shown in Appendix A. Metabolites that were significantly expressed and the fold change of expression levels between asthmatics and controls, and between children with and without mite, food, and IgE sensitization are shown in Table 2. A significantly higher plasma histidine level in parallel with lower urinary 1-methylnicotinamide and trimethylamine N-oxide (TMAO) levels was found in children with asthma compared to the healthy controls. Compared to children without allergic sensitization, 11 (92%) plasma metabolites and 8 (80%) urinary metabolites were found to be significantly different in children with IgE and food sensitization, respectively.

### 3.3. Association between Plasma and Urine Metabolites

The Spearman’s rank correlation coefficients between plasma and urine metabolites are shown in Figure 1. The plasma pyroglutamate, 3-hydroxybutyric acid, and acetone were strongly negatively correlated, whereas plasma propylene glycol was positively correlated, with most of the urine metabolites. Among them, plasma 3-hydroxybutyric acid was strongly correlated with urine 3-methyl-2-oxovaleric acid, 3-hydroxyisovaleric acid, 2-hydroxyisobutyric acid, 3-hydroxyisobutyric acid, and alanine. Furthermore, plasma leucine, tyrosine, and valine also showed strong correlations with urine 3-hydroxyisovaleric acid, 2-hydroxyisobutyric acid, and alanine. For metabolites that existed in both plasma and urine, only creatine and lysine were negatively correlated between plasma and urine (Appendix A).

### 3.4. Association of Atopic Indices Associated Metabolites between Plasma and Urine Metabolome

Appendix A shows the correlations of differentially expressed metabolites associated with atopic indices between the plasma and urine metabolomes. Asthma associated plasma histidine was not correlated with any urine metabolites, whereas urine 1-methylnicotinamide and TMAO showed a strong positive correlation with plasma mannose and lysine, respectively. IgE sensitization associated plasma valine and lysine were significantly negatively correlated with 2-hydroxyisobutyric acid and 3-hydroxyisovaleric acid, respectively (Appendix A). In contrast, food sensitization associated urine 3-methyl-2-oxovaleric acid and 3-hydroxyisobutyric acid showed a significant negative correlation with plasma pyroglutamate and 3-hydroxybutyric acid, respectively (Appendix A).

### 3.5. Metabolites Associated with Atopic Indices Related to Allergen-Specific IgE Levels

The correlations of atopic indices associated plasma and urine metabolites with total serum IgE and allergen-specific IgE levels are shown in Figure 2. Total serum IgE levels were significantly and positively correlated with serum specific IgE levels to *D. pteronyssinus*, *D. farinae*, egg white, and cow’s milk (*p*< 0.01). Among them, a strong correlation was found between *D. pteronyssinus*- and *D. farinae*-specific IgE levels, and between egg white- and cow’s milk-specific IgE levels, but was not found between the mite and food allergen-specific IgE levels. However, plasma metabolites associated with IgE sensitization were not correlated with total serum IgE and allergen-specific IgE levels. In contrast, the majority of urine metabolites associated with food sensitization were significantly and positively correlated with total serum IgE and egg white-specific IgE levels (*p*< 0.05). Furthermore, asthma associated urine TMAO was significantly correlated with lysine and methionine, whereas urine 1-methylnicotinamide was significantly correlated with acetic acid and lactic acid.

### 3.6. Metabolic Pathway and Functional Analysis

The metabolic functional pathways related to allergic sensitization are shown in Table 3. Plasma glycolysis or gluconeogenesis with pyruvate metabolism and urine amino acid metabolism were significantly associated with children with allergic sensitization compared to the healthy controls. Reduced pyruvic acid and lactic acid levels accompanied with increased acetic acid levels were found in plasma pyruvate metabolism. However, increased leucine, valine, and 3-hydroxyisobutyric acid levels, and 3-phenylpropionate, N-phenylacetylglycine, and tyrosine levels were found in the urine valine, leucine, and isoleucine degradation pathway and in phenylalanine metabolism, respectively.

## 4. Discussion

Metabolic profiling and analysis was applied to explore the metabolic effects of human diseases and some metabolites have been successfully identified to be association with asthma. However, different sample types identify the distinct metabolic profile of disease state. This study provides an integrative analysis of plasma and urine approaches for a comprehensive framework of assessing childhood asthma.

Blood circulating through the body carries many small molecules metabolites that are endogenously produced by organisms and exogenous chemicals, including drugs, foods, and environmental allergies. In contrast, urine is generated by the kidneys as they extract the water-soluble waste products from the bloodstream, as well as a variety of other excess compounds [16]. Nonetheless, protein-bound metabolites may not be completely filtered through the glomeruli, and active tubular secretion and reabsorption may also influence urinary levels [17]. This phenomenon, in principle, can explain why metabolites that existed in both plasma and urine were not strongly correlated with each other in this study.

3-Hydroxybutyric acid is synthesized in the liver from acetyl-CoA and is a metabolite of fatty acids and ketogenic amino acids, such as leucine and isoleucine [18]. In this study, strong and significant correlations were observed between the plasma 3-hydroxybutyric acid and excreted urine volumes of the hydroxy acids. Furthermore, these urinary hydroxyl acids were strongly correlated with plasma leucine and valine levels. These findings support the statement that the accumulation of the urinary hydroxy acids during ketoacidosis can represent the state of ketosis [19], which may be caused by the derangement of the metabolism of leucine, isoleucine, and valine.

Total serum and allergen-specific IgE levels are integral to the pathogenesis of allergic disorders. As in previous study, total serum IgE levels were strongly correlated with specific IgE levels to mites and foods [20]. In plasma, however, the majority of differentially expressed metabolites were found in children with IgE sensitization that had reduced pyruvic acid and lactic acid levels. In parallel with the increased acetic acid and valine levels that were observed, plasma pyruvate metabolism, through acetyl-CoA, may play a role in serum IgE production in response to allergies.

In contrast to plasma, the majority of urine metabolites were significantly and differentially expressed in children with food sensitization, and also correlated with total serum IgE levels, supporting that urine metabolome may primarily reflect nutrition status, dietary intake, and allergy outcomes [21,22]. Furthermore, the food sensitization associated urine 3-hydroxyisobutyric acid and 3-methyl-2-oxovaleric acid were an intermediate in the metabolism of valine and isoleucine, respectively [23,24]. An increase in urine levels of valine, leucine, and the hydroxy acids in this study indicate that branched-chain amino acid metabolism (BCAAs; valine, leucine, and isoleucine) may impact allergic responses to food and allergy-related outcomes.

Metabolomics profiling and analysis of urine samples have successfully identified microbial-derived metabolites associated with asthma development [25]. In this study, urine TMAO, a dietary compound produced by bacteria in the intestine [26], was identified to be associated with food sensitization and childhood asthma. Furthermore, urine N-phenylacetylglycine, a gut microbial co-metabolite [27], was strongly correlated with total serum IgE levels and food allergen-specific IgE levels. These microbial-host metabolites result from the transformation of specific dietary components by select microbial species. Thus, the dysbiosis of gut microbiota utilizing diet-dependent metabolic pathways may be key in determining the susceptibility of humans to develop allergic diseases.

Histidine, an essential amino acid, is critical in an enzymatic reaction as it is responsible for the production of histamine, which is well known for its role in acute allergic reactions in asthma cases [28]. In contrast, urine 1-methylnicotinamide, a metabolite of dietary niacin, was strongly associated with asthma and also plasma mannose. Mannose-binding lectin, an important protein in innate immunity, regulates the inflammatory responses after binding mannose and is implicated in asthma by contributing to airway inflammation [29]. 1-Methylnicotinamide is reported to have a protective role against asthma exacerbation, with one possible mechanism being through its anti-inflammatory effects that are related to mannose-binding lectin levels [30].

Metabolomics is useful to identify biomarkers and pathways involved in complex mechanisms regulating asthma. By using NMR profiling, numerous pathways including amino acids (alanine, arginine, phenylalanine, and threonine) [5,31], acid salt (formate, hippurate, and succinate) [5,32], and alcohol (methanol) [5] pathways have been reported to be associated with asthma. Although there is considerable consistency in the metabolites and pathways in these studies, a large number of metabolic pathways are not replicated between biospecimens. This particularly explains the distinct profiles in the blood and urine related to asthma in this study, supporting an integrative metabolomics data analysis for asthma, as a recent comparative metabolomics analysis suggests [33].

The major limitations of this study were the relatively small sample size and the low analytical sensitivity to low-abundance metabolites (< 100 nmol/L) that is present when using ^1^H-NMR spectroscopy. However, ^1^H-NMR spectroscopy is a powerful technique that enables structure determination of molecules in solution with a high analytical reproducibility. Furthermore, an age- and sex-matched design in this study eliminates the dissimilarities in metabolic profiles of subjects. Most importantly, an integrative analysis of the plasma and urine metabolomes provides more comprehensive metabolic information on allergic diseases.

In conclusion, plasma and urine metabolome reflect different metabolic pathways on allergic reactions and outcomes. Plasma pyruvate metabolism to energy and to the amino acids appears to be associated with serum IgE production, whereas urine branched-chain amino acid metabolism primarily reflects food allergic reactions against allergies. Furthermore, urine microbial-host co-metabolites related to food sensitization, such as TMAO and N-phenylacetylglycine, exemplify the impact of gut microbial dysbiosis on allergies. Nonetheless, further functional studies are warranted to investigate these associations more comprehensively.

## Figures and Tables

**Figure 1 jcm-09-00887-f001:**
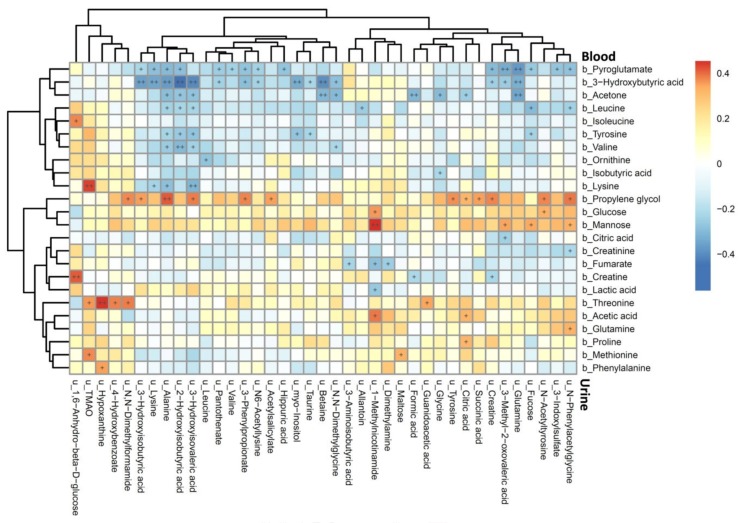
Heatmap of Spearman’s rank correlation coefficients between plasma and urine metabolites. Color intensity represents the magnitude of correlation. Red color = positive correlations; blue color = negative correlations. + symbol means a *p*-value < 0.05; ++ symbol means a *p*-value < 0.01.

**Figure 2 jcm-09-00887-f002:**
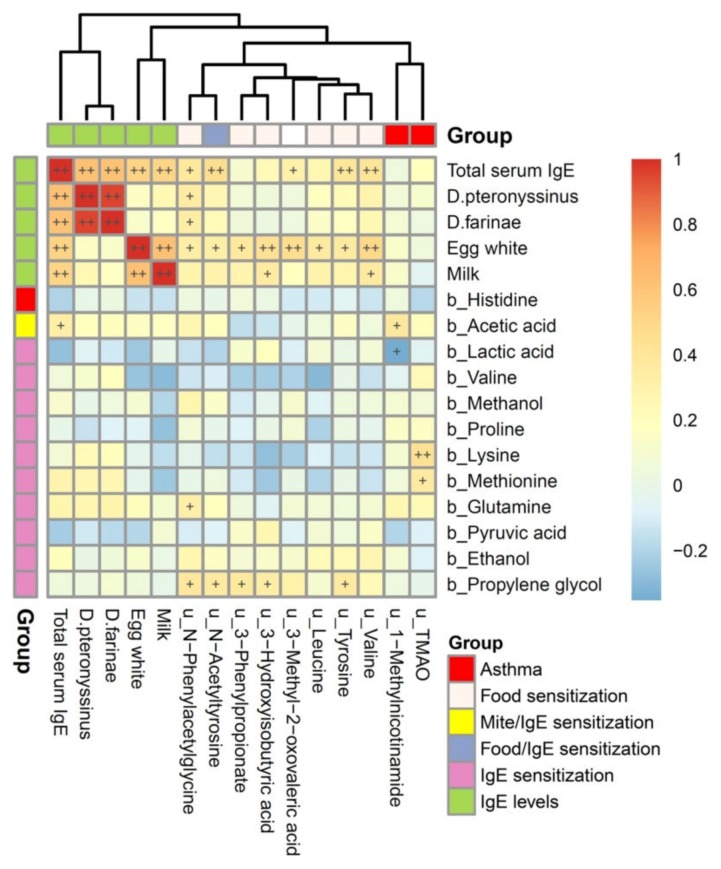
Heatmap of Spearman’s rank correlations of atopic indices associated plasma and urine metabolites with total serum IgE and allergen-specific IgE levels for asthma. Color intensity represents the magnitude of correlation. Red color represents positive correlations; blue color represents negative correlations. + symbol means a *p* -value < 0.05; ++ symbol means a *p* -value < 0.01.

**Table 1 jcm-09-00887-t001:** Epidemiologic characteristics of the 54 children investigated in this study.

Characteristics	Asthmatics (n = 28)	Controls (n = 26)	*p* Value
Age (yr)	3.62 ± 0.66	3.61 ± 0.73	0.971
Sex, male	19 (67.9%)	16 (61.5%)	0.627
Maternal atopy	13 (48.1%)	9 (34.6%)	0.318
Passive smoking	9 (32.1%)	10 (38.5)	0.627
Household income			0.559
Low, ≤ 500,000 NTD	10 (37.0%)	12 (46.2%)	
Medium, 500,000–1,000,000 NTD	11 (40.7%)	11 (42.3%)	
High, > 1,000,000 NTD	6 (22.2%)	3 (11.5%)	
Allergen-specific IgE, kU/L			
*D. pteronyssinus*	29.67 ± 53.07	0.34 ± 0.55	**0.001**
*D. farinae*	14.41 ± 28.04	0.17 ± 0.26	**0.001**
Egg white	0.63 ± 0.83	0.33 ± 0.52	0.203
Cow’s milk	0.58 ± 0.90	0.22 ± 0.27	0.052
Sensitization			
Mite	19 (67.9%)	5 (19.2%)	**<0.001**
Food	15 (53.6%)	7 (26.9%)	**0.046**
IgE > 100 kU/L	14 (50.0%)	2 (7.7%)	**0.001**
Total serum IgE, kU/L	224.00 ± 292.43	47.79 ± 60.67	**0.001**

Data shown are mean ± SD or number (%) of patients as appropriate. yr, year; NTD, New Taiwan Dollar; IgE, immunoglobulin E. All *p* values < 0.05, which is in bold, are significant.

**Table 2 jcm-09-00887-t002:** The VIP score and fold change of metabolites significantly differentially expressed between children with asthma and controls, and between children with and without mite, food and IgE sensitization.

		Asthma			Mite Sensitization			Food Sensitization			IgE Sensitization		
Metabolites	Chemical Shift, ppm	VIP Score *	Fold Change †	*p* ‡	VIP Score	Fold Change	*p*	VIP Score	Fold Change	*p*	VIP Score	Fold Change	*p*
**Plasma**													
Histidine	7.030–7.078 (s)	1.28	1.13	**0.009**	0.83	1.08	0.081	0.04	0.98	0.934	0.36	1.05	0.293
Acetic acid	1.850–1.930 (d)	1.19	1.41	0.226	2.02	1.37	**0.031**	1.04	1.24	0.254	1.68	1.48	**0.009**
Lactic acid	1.304–1.340 (d)	1.02	0.86	0.304	0.40	0.94	0.678	1.38	0.80	0.125	1.41	0.63	**0.030**
Valine	3.588–3.615 (d)	0.45	1.04	0.309	0.69	1.06	0.097	0.12	1.01	0.768	0.64	1.10	**0.027**
Methanol	3.347–3.363 (s)	0.82	1.13	0.241	0.27	1.04	0.686	0.45	1.07	0.479	1.19	1.33	**0.009**
Proline	3.301–3.347 (dt)	0.41	1.04	0.449	0.44	1.03	0.397	0.18	0.98	0.720	0.90	1.15	**0.011**
Lysine	2.999–3.024 (t)	0.02	1.00	0.947	0.36	1.02	0.291	0.12	0.99	0.722	0.38	1.04	**0.036**
Methionine	2.610–2.653 (t)	0.48	1.05	0.330	0.63	1.06	0.186	0.03	1.01	0.941	1.00	1.16	**0.002**
Glutamine	2.416–2.494 (m)	0.15	1.02	0.775	0.23	1.02	0.633	0.61	1.06	0.185	0.68	1.11	**0.041**
Pyruvic acid	2.354–2.376 (s)	0.87	0.86	0.352	0.79	0.87	0.379	0.80	0.89	0.351	1.27	0.66	**0.025**
Ethanol	1.148–1.181 (t)	1.86	1.69	0.062	0.44	1.05	0.652	0.82	0.95	0.383	1.50	1.64	**0.026**
Propylene glycol	1.123–1.148 (d)	1.52	1.80	0.241	0.37	1.26	0.766	0.80	1.47	0.501	2.20	2.65	**0.009**
**Urine**													
1-Methylnicotinamide	8.874–8.906 (d)	2.44	0.64	**0.011**	0.21	1.00	0.818	0.31	1.32	0.737	1.04	0.80	0.289
TMAO	3.256–3.261 (s)	1.74	0.75	**0.025**	0.80	1.15	0.274	0.01	1.04	0.992	1.10	1.29	0.163
N-Phenylacetylglycine	7.330–7.370 (m)	0.05	0.99	0.946	1.01	1.20	0.151	1.39	1.36	**0.024**	0.93	1.22	0.224
Tyrosine	6.874–6.898 (m)	0.11	0.98	0.871	0.46	1.07	0.454	1.45	1.31	**0.006**	0.93	1.23	0.158
3-Phenylpropionate	2.870–2.907 (t)	0.23	0.97	0.691	0.35	1.03	0.519	1.13	1.20	**0.016**	0.13	1.02	0.828
3-Methyl-2-oxovaleric acid	1.083–1.105 (d)	0.25	0.98	0.725	0.26	1.03	0.692	1.68	1.40	**0.003**	0.28	1.07	0.693
3-Hydroxyisobutyric acid	1.050–1.070 (d)	0.03	1.24	0.974	0.88	1.40	0.231	1.48	1.74	**0.022**	1.03	1.72	0.196
Valine	1.023–1.041 (d)	0.18	0.99	0.772	0.80	1.12	0.164	1.64	1.35	**0.001**	0.37	1.07	0.555
Leucine	0.936–0.966 (t)	0.41	0.94	0.490	0.81	1.11	0.132	0.97	1.16	**0.042**	0.06	1.00	0.915
N-Acetyltyrosine	6.841–6.869 (d)	0.38	1.02	0.568	0.00	0.97	1.000	1.16	1.24	**0.030**	1.37	1.25	**0.034**

* VIP score were obtained from PLS-DA. † Fold changes were calculated by dividing the value of metabolites in children with asthma by controls, and in children with by without mite, food and IgE sensitization. ‡ All FDR-adjusted *p* values < 0.05, which is in bold, are significant. VIP, Variable Importance in Projection; IgE; immunoglobulin E; ppm, parts per million; m, multiplet; s, singlet; d, doublet; t, triplet; dt, doublet of triplet.

**Table 3 jcm-09-00887-t003:** Metabolic pathway and function analysis of metabolites associated with allergic sensitization for childhood asthma.

Sample	Metabolites	Pathway Name	Total	Hits	Raw *p*	FDR	Function
Plasma	Histidine, Glutamine, Methionine, Valine, Lysine, Proline	Aminoacyl-tRNA biosynthesis	75	6	5.97E-07	4.77E-05	Genetic Information Processing, Translation
	Ethanol, Pyruvic acid, Lactic acid, Acetic acid	Glycolysis or Gluconeogenesis	31	4	1.04E-05	3.16E-04	Carbohydrate metabolism
	Pyruvic acid, Propylene glycol, Lactic acid, Acetic acid	Pyruvate metabolism	32	4	1.18E-05	3.16E-04	Carbohydrate metabolism
Urine	Leucine, Valine, 3-Hydroxyisobutyric acid	Valine, leucine and isoleucine degradation	40	3	4.71E-04	2.68E-02	Amino acid metabolism
	N-Phenylacetylglycine, Tyrosine, 3-Phenylpropionate	Phenylalanine metabolism	45	3	6.69E-04	2.68E-02	Amino acid metabolism

Total is the total number of compounds in the pathway; the Hits is the actually matched number from the user uploaded data; the Raw *p* is the original *p* value calculated from the enrichment analysis; the FDR is the portion of false positives above the user-specified score threshold. FDR, false discovery rate.

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
