# Peer review of "Metabolomic Analysis Reveals Distinct Profiles in the Plasma and Urine Associated with IgE Reactions in Childhood Asthma"

_jcm, 2020, doi:10.3390/jcm9030887_

Round 1

Reviewer 1 Report

In the present studies, Chih-Yung Chiu et. al., analysed the metabolic responses in blood and urine samples in children with asthma as compared to healthy controls. The authors detected higher plasma histidine levels and decreased lower urinary 1-methylnicotinamide and trimethylamine N-oxide (TMAO) levels in children with compared to healthy controls. Eleven plasma metabolites and eight urinary metabolites were found to be significantly different in children with IgE and food sensitization, respectively. The microbial-host metabolite urine N-phenylacetylglycine was found to be strongly correlated with total serum and food allergen-specific IgE levels. Finally, they found that plasma pyruvate metabolism to acetic acid was associated with serum IgE, whereas urine branched-chain amino acid metabolism was linked to food allergic reactions. Although this is an interesting study that presents novel data describing the metabolic alterations in childhood asthma, there are several issues that need to be addressed to strengthen the findings and support the conclusions proposed.

Major points:

  1. The authors should investigate the inflammatory response in children with asthma and explore possible correlations between cytokines (i.e.IL-4/5/13, TNFa, Il-1b) levels in the serum with plasma and urine metabolites.

  1. Potential correlations with other clinical parameters of asthmatic disease should be also investigated.

  1. The discussion should describe the findings of other metabolomics studies in other asthmatic populations and compare them to the current findings.

  1. Extensive bioinformatics analyses, including comparative analyses with other metabolomics studies, should accompany these findings.

Minor comments:

  1. The Introduction should clearly present the hypothesis and aims of the present study.
  2. The manuscript should be revised by an English-native person.

Author Response

Reviewer: 1

In the present studies, Chih-Yung Chiu et. al., analysed the metabolic responses in blood and urine samples in children with asthma as compared to healthy controls. The authors detected higher plasma histidine levels and decreased lower urinary 1-methylnicotinamide and trimethylamine N-oxide (TMAO) levels in children with compared to healthy controls. Eleven plasma metabolites and eight urinary metabolites were found to be significantly different in children with IgE and food sensitization, respectively. The microbial-host metabolite urine N-phenylacetylglycine was found to be strongly correlated with total serum and food allergen-specific IgE levels. Finally, they found that plasma pyruvate metabolism to acetic acid was associated with serum IgE, whereas urine branched chain amino acid metabolism was linked to food allergic reactions. Although this is an interesting study that presents novel data describing the metabolic alterations in childhood asthma, there are several issues that need to be addressed to strengthen the findings and support the conclusions proposed.

Major points:

1.The authors should investigate the inflammatory response in children with asthma and explore possible correlations between cytokines (i.e.IL-4/5/13, TNFa, Il-1b) levels in the serum with plasma and urine metabolites.

Ans: Thanks so much for your precious suggestion. Honestly, we have ever tried to link the association between IgE levels and Treg related cytokine including interleukin-10 (IL-10) and transforming growth factor-β (TGF-β), and T-helper 1 (Th1) (interferon-γ [IFN-γ]) and Th2 (interleukin-4 [IL-4] and IL-5) cytokines. Unfortunately, the measured levels of Treg, Th1 and Th2 cytokines were low and close to limit of detection in serum. We therefore measured Th1- and Th2-like chemokines instead of measuring cytokines. Th1-associated CXC chemokine ligand (CXCL10 and CXCL11) and the Th2- associated CC chemokine ligand (CCL17 and CCL22) were measured and their relevance to IgE levels was firstly examined. As you can see the strong correlations between allergen-specific IgE levels, there were strong correlations between these chemokines. However, there were no correlations between IgE levels and chemokines (see the Figure below). Also, there were only a few non-specific plasma metabolites correlated with these chemokines. Our preliminary findings suggest that Th1/Th2 chemokines may not completely represent the Th1/Th2 cytokine immune network linking to allergic reactions for asthma. Also, a low analytical sensitivity to low-abundance metabolites (< 100 nmol/L) using 1H-NMR spectroscopy in this study may be not yet comprehensive enough for analysis. Actually, it may be valuable to investigate the relationship of metabolic profiles with cytokine-chemokine immune network and childhood asthma using liquid chromatography–mass spectrometry (LC-MS). However, by using NMR profiling, there is no exciting results to report at this moment.

2.Potential correlations with other clinical parameters of asthmatic disease should be also investigated.

Ans: In this study, clinical parameters of asthmatic disease have shown in Table 1. As you can see in the table, atopic indices including total serum IgE levels and allergen-specific IgE levels to mite and food were significantly higher in children with asthma than in the healthy controls (P = 0.001). By contrast, there were no differences in other clinical parameters including sex, maternal age, maternal atopy, passive smoking, and household income. Basically, the efforts of the wide metabolomics analysis among allergen sensitization, plasma and urine metabolic profiles in this study can represent enough information to identify metabolic pathways involved in mediating the allergic responses related to asthma.

3.The discussion should describe the findings of other metabolomics studies in other asthmatic populations and compare them to the current findings.

Ans: As reviewer’s suggestion, information regarding the diverse findings of other metabolomics studies and the distinct profiles in the blood and urine of asthma in our study has added and shown below:

In the Discussion,

“Metabolomics is useful to identify biomarkers and pathways involved in complex mechanisms regulating asthma. By using NMR profiling, numerous pathways including amino acids (alanine, arginine, phenylalanine, and threonine) [5,31], acid salt (formate, hippurate, and succinate) [5,32], and alcohol (methanol) [5] pathways have been reported to be associated with asthma. Although there is considerable consistency in the metabolites and pathways in these studies, a large number of metabolic pathways are not replicated between biospecimens. This particularly explains the distinct profiles in the blood and urine related to asthma in this study, supporting an integrative metabolomics data analysis for asthma, as a recent comparative metabolomics analysis suggests [33].”

4.Extensive bioinformatics analyses, including comparative analyses with other metabolomics studies, should accompany these findings.

Ans: Thanks so much for your precious suggestion. It is definitely a good idea to have an extensive bioinformatics analysis including comparative analyses with other metabolomics studies in this study. Honestly, it will be a totally different way of studying. Luckily, a comparative analysis of asthma metabolomics studies has recently published. Considering the main hypothesis and major goal in this study, major information of this new published analysis has added, cited, and shown below:

In the Discussion,

“Metabolomics is useful to identify biomarkers and pathways involved in complex mechanisms regulating asthma. By using NMR profiling, numerous pathways including amino acids (alanine, arginine, phenylalanine, and threonine) [5,31], acid salt (formate, hippurate, and succinate) [5,32], and alcohol (methanol) [5] pathways have been reported to be associated with asthma. Although there is considerable consistency in the metabolites and pathways in these studies, a large number of metabolic pathways are not replicated between biospecimens. This particularly explains the distinct profiles in the blood and urine related to asthma in this study, supporting an integrative metabolomics data analysis for asthma, as a recent comparative metabolomics analysis suggests [33].”

Minor comments:

1.The Introduction should clearly present the hypothesis and aims of the present study.

Ans: As reviewer’s suggestion, the hypothesis and major goal of the present study have clearly presented as below:

In the Introduction,

“………………We hypothesized that integrated analyses of metabolic profiles in blood and urine would provide more detailed information, highlighting the complexity of childhood asthma. Furthermore, a comprehensive understanding of the complexities of metabolic pathways involved in asthma may determine potential pathways involved in mediating the allergic responses to allergen and identify metabolite biomarkers potentially important for the treatment of asthma. The major aim of this study was to determine the plasma and urine metabolic profiles using 1H-NMR spectroscopy in patients with childhood asthma and healthy controls. The relationship between metabolites and atopic indices including IgE and allergen sensitization were assessed, and their relevance to the risk of asthma was examined.”

2.The manuscript should be revised by an English-native person.

Ans: As reviewer’s suggestion, the text of this manuscript has edited for grammar, spelling and other errors by the language professionals. Please see attached certificate of English editing.

Thanks for your precious comments and suggestions. Hope these responses could answer all your questions.

Thanks again.

Sincerely

Chih-Yung Chiu MD. PhD.

Department of Pediatrics, Chang Gung Memorial Hospital at Linkou

5, Fuxing St., Guishan Dist., Taoyuan, Taiwan

Tel: 886-3-3281200 ext 8202; E-mail: [email protected]

References in response to reviewers:

[33]. Kelly, R.S.; Dahlin, A.; McGeachie, M.J.; Qiu, W.; Sordillo, J.; Wan, E.S.; Wu, A.C.; Lasky-Su, J. Asthma Metabolomics and the Potential for Integrative Omics in Research and the Clinic. Chest 2017, 151, 262-277.

Reviewer 2 Report

Only if it is possible please extend the introduction section

Author Response

Ms. Ref. No.: jcm-738215

Title: Metabolomic Analysis Reveals Distinct Profiles in the Plasma and Urine Associated with IgE Reactions in Childhood Asthma

Journal of Clinical Medicine

Reviewer: 2

COMMENTS FOR THE AUTHOR:

  1. Only if it is possible please extend the introduction section

Ans: Thank you so much for the positive feedback. As reviewer’s suggestion, the introduction section regarding the hypothesis and major goal of the present study has extended as below:

In the Introduction,

“………………We hypothesized that integrated analyses of metabolic profiles in blood and urine would provide more detailed information, highlighting the complexity of childhood asthma. Furthermore, a comprehensive understanding of the complexities of metabolic pathways involved in asthma may determine potential pathways involved in mediating the allergic responses to allergen and identify metabolite biomarkers potentially important for the treatment of asthma. The major aim of this study was to determine the plasma and urine metabolic profiles using 1H-NMR spectroscopy in patients with childhood asthma and healthy controls. The relationship between metabolites and atopic indices including IgE and allergen sensitization were assessed, and their relevance to the risk of asthma was examined.”

Thanks for your precious comments and suggestions. Hope these responses could answer all your questions.

Thanks again.

Sincerely

Chih-Yung Chiu MD. PhD.

Department of Pediatrics, Chang Gung Memorial Hospital at Linkou

5, Fuxing St., Guishan Dist., Taoyuan, Taiwan

Tel: 886-3-3281200 ext 8202; E-mail: [email protected]

Round 2

Reviewer 1 Report

The manuscript has been improved and is of sufficient quality for publication.